# The Whole-Exome Sequencing of a Cohort of 19 Families with Adolescent Idiopathic Scoliosis (AIS): Candidate Pathways

**DOI:** 10.3390/genes14112094

**Published:** 2023-11-17

**Authors:** Laura Marie-Hardy, Thomas Courtin, Hugues Pascal-Moussellard, Serge Zakine, Alexis Brice

**Affiliations:** 1Brain Institute of Paris, 43-87 bd de l’Hôpital, 75013 Paris, France; 2Clinique Maussins Nollet, Ramsay Génerale de Santé, 67 Rue de Romainville, 75019 Paris, France; serge.zakine@gmail.com

**Keywords:** idiopathic scoliosis, exome sequencing, myosin, actin, calmodulin

## Abstract

A significant genetic involvement has been known for decades to exist in adolescent idiopathic scoliosis (AIS), a spine deformity affecting 1–3% of the world population. However, though biomechanical and endocrinological theories have emerged, no clear pathophysiological explanation has been found. Data from the whole-exome sequencing performed on 113 individuals in 19 multi-generational families with AIS have been filtered and analyzed via interaction pathways and functional category analysis (Varaft, Bingo and Panther). The subsequent list of 2566 variants has been compared to the variants already described in the literature, with an 18% match rate. The familial analysis in two families reveals mutations in the BICD2 gene, supporting the involvement of the muscular system in AIS etiology. The cellular component analysis revealed significant enrichment in myosin-related and neuronal activity-related categories. All together, these results reinforce the suspected role of the neuronal and muscular systems, highlighting the calmodulin pathway and suggesting a role of DNA-binding activities in AIS physiopathology.

## 1. Introduction

Adolescent idiopathic scoliosis (AIS) is the most common spinal deformity, affecting 1–3% of the worldwide population [1]. It is defined as a three-dimensional spinal deformity with curves exceeding 10° of magnitude in the coronal plane [2]. The medical management of AIS consists of screening, bracing and surgical correction for curves progressing over 40° despite bracing [3,4]. AIS has a long-term impact on patients’ quality of life and may lead to significant disability [5]. Moreover, up to 1–10% of AIS patients will require corrective surgery with a cost of approximatively USD 50,000, resulting in a heavy financial burden on society [6]. For decades, researchers have tried to understand the etiopathology of scoliosis in order to find treatment options that are not only symptomatic. The role of polygenic risk factors has been supported by the work of Riseborough et al. in 1973, finding a 12% recurrence risk of AIS for first-degree relatives [7]. The two main hypotheses that emerged to explain AIS were the biomechanical/musculoskeletal theory and the hormonal theory (related to estrogen and melatonin) [8]. The biomechanical/musculoskeletal theory hypotheses that the cartilage, bone and/or muscle are subject to disequilibrium during growth, resulting in the three-dimensional deformity of the spine. FBN1 (fibrillin-1, a component of the extracellular matrix), MATN1 (matrilin-1, a cartilage protein) and LBX1 (ladybird homeaobox-1, a regulator of muscle precursor cell migration) have been identified as the most promising candidate genes related to the musculoskeletal theory [9,10,11]. The hormonal theory states that a dysfunction in hormones regulating bone formation (such as estrogen), promoting osteoblast proliferation (such as melatonin) or regulating bone formation (such as leptin) may also induce disequilibrium in spine growth, resulting in AIS. Variants in CALM1 and ESR1 have been found to be associated with AIS in previous studies [10,12]. The recent progress in the field of genetics has allowed family-based studies and GWAS (Genome-Wide Association Study) in scoliosis cohorts. New variants have been identified, and a third hypothesis related to the nervous system has emerged, identifying new candidate genes such as CELSR2, TTL1 or CFAP^298^ [12,13,14,15]. Those works suggest that impairment in processes involving cerebellum growth, axonal migration, or cilium function result in AIS. However, these variants have failed to replicate in other cohorts and none of them seem to explain AIS as a monogenic disease [7,16,17,18]. Indeed, recent reports suggest that the pathophysiology of AIS seems to combine several phenomena involving the muscular and neural systems, regulated by complex hormonal and epigenetic phenomena. Subsequently, studies have tried to identify the pathways and functional categories that may be useful for explaining this highly complex polygenic disease [7,19,20,21]. These studies revealed enrichment in extra-cellular matrices’ genes and actin-based tubular projections and provide a promising lead for the future understanding of AIS.

The main goal of this study was to find new candidate pathways and functional categories involved in scoliosis by analyzing a large cohort of AIS families. 

## 2. Materials and Methods

Patients: As a prospective data collection cohort, nineteen French families with a suppositively dominant inheritance of AIS were enrolled. The inclusion criteria consisted of at least two cases of idiopathic scoliosis in two successive generations. For any affected individual, the participation of at least one sibling without AIS as a healthy relative and both parents were sought ought. All patients underwent an orthopedic examination to confirm the idiopathic nature of the scoliosis (no neurological or syndromic pathology was associated, no dysraphism, normal abdominal reflexes, normal motor function, normal reflexes, absence of vertebral malformation on the radiographic analysis, concordant age of onset, or concordant rate of progression) [22,23]. The radiographic diagnostic of scoliosis relied on the presence of a curve greater than 10° with vertebral rotation [1]. For all subjects, age, gender and medical history was recorded. Additionally, for all AIS patients, magnitude of the main curve and contra-curve (in °) and gibbosity (in cm) were measured, and clinical data associated with AIS, such as age of onset and treatment history (bracing, surgery) were recorded. The pedigree charts of all families were performed on CeGat Software (Version 2.17). This study received ethical committee approval number 2020.02.05 bis_20.01.23.63418 (Comité de protection des personnes Sud-Méditerranée III). All procedures involving human participants were performed in accordance with the 1964 Declaration of Helsinki and its later amendments.

### Methods

AIS patients and genetically relevant healthy relatives were sampled for blood or saliva, and anonymization was obtained by the numerical coding of the identity of the patients. DNA was obtained from 63 affected individuals with AIS and 50 non-affected relatives. DNA was extracted to a final quantity of 100 ng (35 µL at 3 ng·µL^−1^), with quality controls made by gel electrophoresis. Whole Exome Sequencing (WES) was performed on each sample, using the Nextera Rapid Capture MedExome v1.2 kit with paired-end 150-bp sequencing (Illumina NextSeq 500, San Diego, CA, USA), and realigned to the hg38 reference genome using the Burrows-Wheeler Aligner (BWA). The variants were annotated using VARAFT version 2.17 (Université Aix-Marseille, Marseille, France). The genetic variants of the 63 affected subjects were filtered by VARAFT to keep only variants of potential significance. The filtering criterions were stringent: exonic mutation (non-synonymous SNV, frameshift deletion or insertion, stop-gain), CADD score > 20 and frequency < 0.001. The results of the familial analysis of the cohort (affected patient versus healthy relatives, by family) are still ongoing and will be presented in a following report; functional validation of variants are still in progress, except for two interesting variants in one gene found in two families.
Literature comparison: A systematic analysis of the literature was performed on PubMed with the MeSH terms “idiopathic scoliosis” and “variant” to collect a list of genes significantly associated with AIS in clinical cohorts with confirmed genetic testing. Matches between this list and the list of SCOGEN variants were searched using ExcelStat.Interaction pathways analysis: The list of variants obtained was then submitted to GO-enrichment analysis using BiNGO (Biological Networks Gene Ontology tool) Cytoscape plugin. The data were analyzed using an overrepresentation binomial test, and *p*-values were corrected via Benjamini-Hochberg false discovery rate (FDR). Significance level was set at *p* < 0.05.Functional categories analysis: In order to identify the functional categories to which the variants present in our patients with AIS belonged, an analysis with Panther Version 18.0, (available at http://pantherdb.org/, accessed on 1 May 2023) was conducted. To retain only the most meaningful variants, the SCOGEN list was reduced to those present in 3 or more individuals. This subsequent list of 490 genes’ variants was then compared using a statistical overrepresentation test (binomial testing) to the GO cellular component list with the reference list on homo sapiens’ genome.

## 3. Results

### 3.1. Patients

The cohort comprised 19 families, consisting of 63 AIS patients and 50 healthy relatives, amounting to 113 individuals in total. The families’ pedigrees are displayed in Figure 1.

All 63 patients presented idiopathic scoliosis, with a mean Cobb angle for the cohort of 28° ± 8. Among the 63 patients, 21 (34%) underwent corrective surgery and 30 (48%) underwent bracing. Forty-nine patients with AIS were female (78%), whereas fourteen (22%) were male. All families were constituted of French citizens.

### 3.2. Variants

The initial analysis of the dataset of the 63 affected individuals resulted in a list of 2566 variants (Appendix A) that passed the filtering criterion of the VARAFT analysis. In total, 756 (67%) of the variant’s genes were unique, 160 (14%) occurred twice and 216 (19%) occurred three times or more. Most variants were missense (Figure 2). 

Comparison to the literature

The literature search of the terms “idiopathic scoliosis” and “variant” on PubMed retrieved 131 studies, of which 25 were excluded, being off-topic. Out of the 106 remaining studies, a list of 114 genes of interest was extracted. The comparison of the SCOGEN datalist to the literature list found 19 matches (18%) between the two datasets. The results are summarized in Table 1.

BINGO analysis

The 409 genes selected in the SCOGEN datalist were analyzed using BINGO to decipher their molecular function. Twenty-three clusters with a corrected *p*-value < 1 × 10^−3^ were identified. These results are presented in Table 2. The first nine clusters in terms of *p*-value were linked to binding activities, notably to nucleosides, nucleotides, proteins and calmodulin. 

A visual representation of these results is also displayed in Figure 3.

Panther analysis

The goal of this analysis was to reveal the most significant cellular component location of the genes of interest, represented at least three times in the SCOGEN list, according to the GO annotation system. The most significant results (according to the corrected *p*-value (False Discovery Rate)) are presented in Table 3. 

Myosin-related categories were represented in three of the top ten categories sorted by *p*-value, such as the myosin II complex (15.7-fold enrichment), myosin complex (8.6-fold enrichment) and muscle myosin complex (19.6-fold enrichment). The neuronal network was the second-most represented system with neuron-projection (2.01-fold enrichment), dendrite (2.35-fold enrichment), dendritic tree (6.4-fold enrichment), and somatodendritic compartment (8.79-fold enrichment).

### 3.3. Familial Analysis—BICD2 Gene

The familial analysis, looking for relevant variants present in at least 75% of the affected individuals in one family and absent in the healthy relatives, highlighted the presence of a variant in BICD2 (BICD cargo adaptor 2) in the family SCOGEN017 (Figure 4). This variant was present in four out of the five scoliosis patients and absent in the healthy relative. This mutation is a non-synonymous SNV (single nucleotide variation), occurring on chromosome 9, position 92722745, on the BICD2 gene, exon3: c.517C>G. This mutation is located in the CC1 zone (coiled-coil 1) of the protein in the dynein-dynactin complex. The predictive scores of this variant are 32 for the CADD score and 0.999 for the DANN score. This mutation is predicted to damage the protein: Provean predictive score: −6.51 (Protein Variation Effect Analyser, available at: https://www.jcvi.org/research/provean, accessed on 1 May 2023). Additionally, another variant of interest located on the BICD2 gene was found in another family: SCOGEN018. This variant, also a non-synonymous SNV, c.1069C>T (rs202119238), located in the CC2 zone (kinesin interaction), was present in 2/2 of the affected individuals (females) but also on one healthy relative (male) (Figure 4). The associated predictive scores were CADD 30, DANN: 0.999 and Provean −4.6.

Those variants were considered to be relevant to AIS, due to both their distribution in the families, especially in SCOGEN017, but also to the physiopathology of BICD2. The absence of phenotype in the male subject in SCOGEN018 family that presented the same variant as his affected mother and sister might be explained by the Carter effect, as described in AIS by Kruse et al. [24]. BICD2 is a motor-adaptor protein implicated in the dynein-dynactin motor complex within the muscular system which also interacts with microtubules [25]. In human pathology, mutations of BICD2 have been found to be responsible for SMA (Spinal Muscular Amyotrophy) in several studies [26,27,28]. The phenotype varies from the pauci-symptomatic form to severe ones in neonatal mortality [26,29]. The patients in SCOGEN017 and SCOGEN018 did not clinically present any sign of muscle- or second motoneuron impairment (no fasciculation, amyotrophia or motor deficit). They did not consent to an additional electromyogram to the study protocol, unfortunately. We may hypothesize that those BICD2 mutations might be responsible for the AIS phenotype either via sub-clinical forms of muscular impairment or via their link to the microtubules, as suggested by Baschal et al. [19].

## 4. Discussion

This study provides a comprehensive description of the rare genetic variants found in 409 genes in a large cohort of familial AIS. The analysis of their cellular function highlights an important number of genes involved in binding activity, suggesting a highly connected network leading to the pathophysiology of AIS. The analysis of cellular component categories reveals a hyper-representation of several myosin- and neuronal-related categories.

The 18% concordance with genes already reported in the literature is encouraging given the heterogeneity of the cohorts in terms of geographical origin (mostly North America and China) and selection criteria (familial studies and GWAS, with or without functional validation). However, several of the most significant genes with variants found in AIS, such as MATN1 and LBX1, were not observed in this cohort [30]. This rate of replication is an echo of the replication studies conducted in Chinese and Canadian cohorts of the ScoliScore designed in an American population, for which only 0–2 of the 53 SNP have significantly replicated in other cohorts [31,32,33]. This genetic scoring system had shown encouraging early results in predicting curve progression on a geographically limited cohort, but results failed to replicate broadly. The results of this cohort are in favor of a heterogenicity of genetic causation in AIS, which may vary between populations. The multiplication of similar studies in different geographic populations of AIS patients might therefore be helpful for understanding AIS etiology broadly.

Amongst the molecular function categories organized in clusters in the datalist retrieved, some referred to functions that may seem ubiquitous. Indeed, eight out of ten of the most represented categories represent nucleoside or nucleotide binding activities. However, the possible involvement of DNA-binding genes has been scarcely described in the literature, except for CHD7 (a DNA binding activity), a gene associated with AIS [34]. One other interesting finding of this study is the representation of “calmodulin-binding” as a molecular function cluster, with a frequency of 2.5% and a *p*-value of 1.5 × 10^–4^. Calmodulin is a calcium-binding receptor protein, involved in muscular contraction and interactions with melatonin—a suspected factor in AIS physiopathology. Calmodulin has been studied in AIS patients and via animal models. Acaroglu et al. have shown that the injection of tamoxifen (a calmodulin antagonist) may influence scoliosis curve regression in pinealectomized chickens, a scoliosis model [35]. The more noticeable clinical studies on AIS patients and calmodulin were conducted by Lowe et al. [36], who discovered that platelet calmodulin levels had then been sampled in AIS patients and correlate closely with curve progression, and by Marcucio et al. and Zhao et al. [37,38], who both found different levels of calmodulin in the paraspinal muscles (convex and concave sides) of AIS patients. CALM-1 (calmodulin 1) variants have been found to be associated with AIS in several cohorts [10,39,40]. In addition, the melatonin-calmodulin interactions involve cytoskeleton, filamentous actin and microtubule assembly, which have been shown by Miller et al. to be key factors in AIS physiopathology [19,41,42].

Panther analysis of the SCOGEN datalist reveals a significant enrichment in several GO cellular components such as myosin-related categories (myosin II complex, myosin complex and muscle myosin complex), plasma membrane, neuronal network (neuron projection, dendrite and dendritic tree) and stereocilium categories. The enrichment in the stereocilium category confirms the findings of Terhune et al., who reported significant cytoskeletal variant enrichment in a comparable cohort [7]. The enrichment in myosin found in the SCOGEN analysis, combined with the results of Baschal et al. in a similar work highlighting actin-based projection enrichment, suggests a clear involvement of the muscular complex, with possible variations amongst series that may be explained by geographical differences between studied populations [19]. Those results altogether may suggest that the binding activities impaired in AIS are predominantly located at the neuro-muscular junction and/or in muscles. The finding of two variants on a candidate gene BICD2 in the familial analysis of this cohort also re-enforces that hypothesis. The BICD2 protein is involved in dynein and dynactin interactions in active motor complexes but also plays a role in neural tissue development. Additionally, a novel mutation in BICD2 has recently been described as causative for a Cohen-like syndrome, with neurological impairment and amyotrophy [43]. BICD2 mutations have been shown to increase microtubules’ stability in motor neurons leading to axonal aberrations [44]. Interestingly, several studies have highlighted the role of microtubules, and especially stereocilium and motile cilium in AIS etiopathology [19,45]. Could AIS be the result of the impairment of several connections at the neuro-muscular junction? These hypotheses are still yet to be confirmed and the answer might result in a more complex model involving genetic variants as described in this study, combined with epigenetic factors [46,47].

## 5. Conclusions

This analysis of a large cohort of idiopathic scoliosis families confirmed the involvement of myosin, calmodulin and genes related to the neural system in AIS etiology, all sophisticatedly intricated with multiple interactions via protein bindings. Two candidate variants for AIS in the BICD2 gene were uncovered, re-enforcing the role of the muscular system and the microtubules in AIS physiopathology.

## Figures and Tables

**Figure 1 genes-14-02094-f001:**
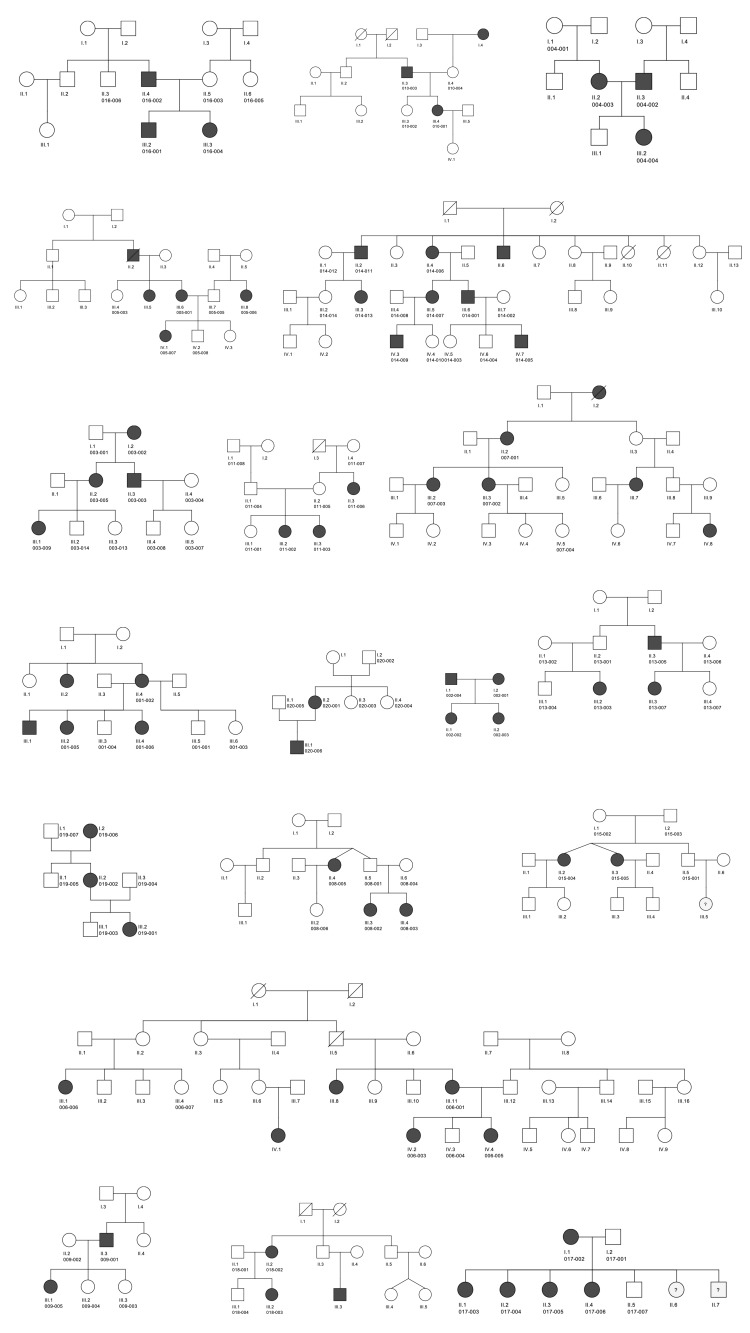
Pedigrees of the 19 families of the SCOGEN cohort. The pedigrees were created on CeGat Software. Patients with confirmed AIS are represented in black and healthy relatives in white.

**Figure 2 genes-14-02094-f002:**
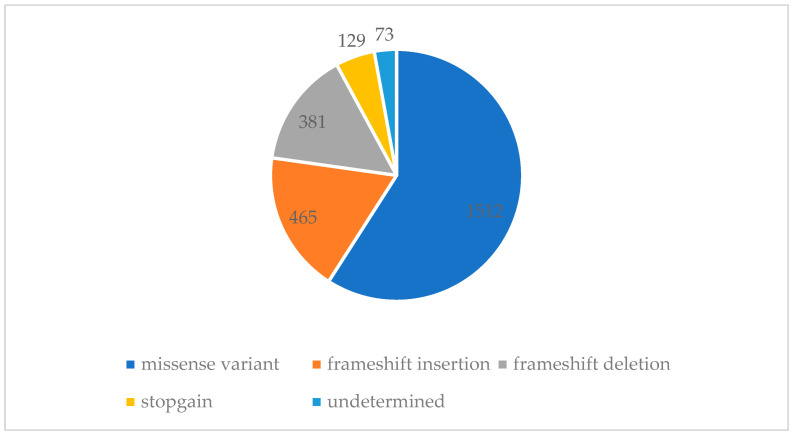
Pie-chart representing the repartition of category types of the variants found in the SCOGEN list. SNV: Single nucleotide variation. Most of the variants (58%) were missense variants and frameshift insertions (18%).

**Figure 3 genes-14-02094-f003:**
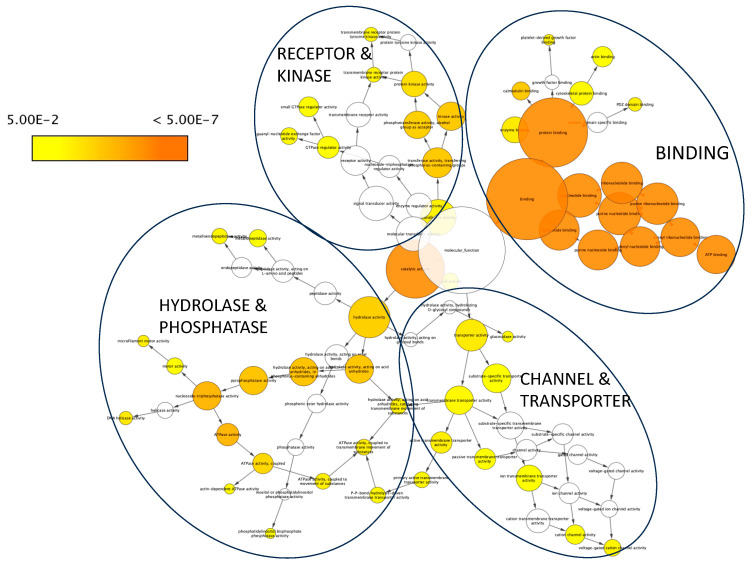
BiNGO visualization of molecular function GO terms in the SCOGEN list of genes. The most significant GO terms are represented in orange (see legend bar with gradient of color according to significance). The size of the node is proportional to the number of genes identified in the SCOGEN list related to this GO term. For easier visualization, the nodes have been pooled into four categories: binding (which is predominantly represented), receptor and kinase, hydrolase and phosphatase, and channel and transporter.

**Figure 4 genes-14-02094-f004:**
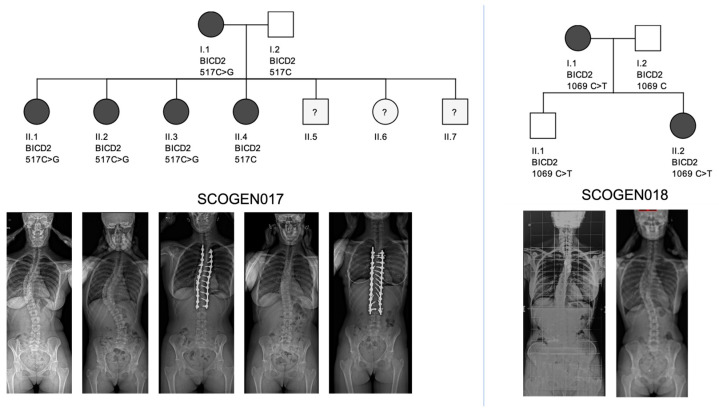
Pedigrees of the SCOGEN017 and SCOGEN018 families regarding mutation status for the two variants in BICD2. In the SCOGEN017 family, patient I.1 had a 35° curve, treated by brace. Patients II.2 and II.4 underwent surgical correction of the scoliosis. Patient II.1 and II.4 were treated by bracing with curves of 20 and 30°, respectively. In the SCOGEN018 family, patients I.1 and II.2 were treated by bracing for thoracic curves of 25 and 35°. All main curves in both families were right thoracic curves, with an age of onset of the scoliosis between 11–15 years old.

**Table 1 genes-14-02094-t001:** List of the 114 genes of interests associated with AIS in the literature, sorted into two categories, depending on their occurrence in the SCOGEN list of genes.

**Found in SCOGEN datalist**	ABCC1- AJAP1- CHD7- COL11A1- DLG1- DST- EPHA7- EPHB6- LRRK2- MACF1- MAGI1- MCM3- MTNR1A- NTF3- PAX3- PCDHA11- SEMA4C- TNIK- VANGL1
**Not found in SCOGEN datalist**	ABO- ADAMTSL2- ADGRG6- ADIPOQ- AKAP2- AKAP9- AQP2- ATP12A- ATRN- BLC2- BMP4- BNC2- BOC- C2CD3- CALM1- CCKBR- CDH13- CELSR2- CEP290- CHI3L1- COL11A2- COMP- CPEB1- CSF3R- DNAAF1- DNAH6- DNAH8- DNHD1- EGR1- EPHA4- EPS8- ESR1- ESR2- FAT3- FBN1- FBN2- FGFR1- FLRT2- FNLB- GJB4- GNGT2- GPR126- GPR50- GRID2- HGF- HHIP- IGF1- IL17RC- IL6- KCNJ2- KIF15- KIF6- LBX1- LEP- LEPR- LRP2- LTBP4- MAPK7- MATN1- MEIS1- MGA- MMP3- MTNR1B- NEDD4- NPHP4- NPY4R- NUCKS1- PAX1- PCDHA4- PITX1- POC5- PRMT5- PTK7- PTPRB- ROBO3- SCNN1A- SCO- SEC16B- SELPLG- SLC22A14- SLC26A7- SLC39A8- SNTG1- SOCS3- SOX6- SOX9- STAB1- SULT1C2- TBX6- TGFB1- TNFRSF10C- TPH1- TTLL11- ZMYND10

**Table 2 genes-14-02094-t002:** Corrected *p*-values (Benjamini-Hochberg false discovery rate) and cluster frequencies of the most represented categories of molecular function (overrepresentation binomial test) according to the GO categories. The analysis was created on BINGo.

GO ID	GO Description	Corrected *p*-Value	Cluster Frequency
1882	nucleoside binding	*6.6291 × 10^−12^*	^179/9^70 18.4%
1883	purine nucleoside binding	*6.6291 × 10^−12^*	^178/9^70 18.3%
17076	purine nucleotide binding	*6.6291 × 10^−12^*	^203/9^70 20.9%
30554	adenyl nucleotide binding	*9.4134 × 10^−12^*	^174/9^70 17.9%
32553	ribonucleotide binding	*1.7312 × 10^−11^*	^194/9^70 20.0%
32555	purine ribonucleotide binding	*1.7312 × 10^−11^*	^194/9^70 20.0%
32559	adenyl ribonucleotide binding	*3.0105 × 10^−11^*	^165/9^70 17.0%
5524	ATP binding	*1.1851 × 10^−10^*	^161/9^70 16.5%
166	nucleotide binding	*2.1163 × 10^−9^*	^217/9^70 22.3%
3824	catalytic activity	*9.1279 × 10^−9^*	^413/9^70 42.5%
5488	binding	*1.2959 × 10^−7^*	^846/9^70 87.2%
5515	protein binding	*1.3637 × 10^−7^*	^600/9^70 61.8%
16887	ATPase activity	*1.9803 × 10^−5^*	^44/9^70 4.5%
17111	nucleoside-triphosphatase activity	*1.9803 × 10^−5^*	^79/9^70 8.1%
16462	pyrophosphatase activity	*8.5471 × 10^−5^*	^79/9^70 8.1%
16818	hydrolase activity, acting on acid anhydrides, in phosphorus-containing anhydrides	*9.4250 × 10^−5^*	^79/9^70 8.1%
16817	hydrolase activity, acting on acid anhydrides	*1.0988 × 10^−4^*	^79/9^70 8.1%
5516	calmodulin binding	*1.4874 × 10^−4^*	^25/9^70 2.5%
16772	transferase activity, transferring phosphorus-containing groups	*1.4972 × 10^−4^*	^90/9^70 9.2%
16301	kinase activity	*1.4972 × 10^−4^*	^79/9^70 8.1%
16787	hydrolase activity	*2.7345 × 10^−4^*	^190/9^70 19.5%
16773	phosphotransferase activity, alcohol group as acceptor	*2.7345 × 10^−4^*	^73/9^70 7.5%
42623	ATPase activity, coupled	*6.2897 × 10^−4^*	^35/9^70 3.6%

**Table 3 genes-14-02094-t003:** Occurrences of the variants in the reference genome of Panther, and in the SCOGEN datalist of genes according to their GO cellular component categories. The number of expected occurrences and the corresponding fold-enrichments with the *p*-values are also presented.

GO Cellular Component	*Homo sapiens*	SCOGEN	Expected	Fold-Enrichment	*Corrected p Value*
plasma membrane region	1307	30	13.33	2.25	9.53 × 10^−3^
apical part of cell	461	15	4.70	3.19	1.07 × 10^−2^
myosin II complex	25	4	0.25	15.69	1.08 × 10^−2^
apical plasma membrane	396	13	4.04	3.22	1.24 × 10^−2^
myosin complex	57	5	0.58	8.60	1.26 × 10^−2^
neuron projection	1366	28	13.93	2.01	1.31 × 10^−2^
muscle myosin complex	15	3	0.15	19.61	1.51 × 10^−2^
specific granule	159	7	1.62	4.32	1.57 × 10^−2^
cluster of actin-based cell projections	166	7	1.69	4.13	1.68 × 10^−2^
cytosol	5515	76	56.25	1.35	2.03 × 10^−2^
myosin filament	24	3	0.24	12.26	2.13 × 10^−2^
dendrite	625	15	6.37	2.35	2.17 × 10^−2^
plasma membrane	5909	80	60.27	1.33	2.24 × 10^−2^
dendritic tree	627	15	6.40	2.35	2.29 × 10^−2^
cell periphery	6395	85	65.23	1.30	2.31 × 10^−2^
intracellular anatomical structure	14,910	170	152.08	1.12	2.48 × 10^−2^
stereocilium	56	4	0.57	7.00	2.59 × 10^−2^
vesicle tethering complex	58	4	0.59	6.76	2.86 × 10^−2^
somatodendritic compartment	862	18	8.79	2.05	3.13 × 10^−2^

## Data Availability

Data are available as Appendix A.

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
