# Peer review of "The Whole-Exome Sequencing of a Cohort of 19 Families with Adolescent Idiopathic Scoliosis (AIS): Candidate Pathways"

_genes, 2023, doi:10.3390/genes14112094_

Round 1
Reviewer 1 Report
Comments and Suggestions for Authors
In this article by Laura Marie-hardy et al., they used whole exome sequencing in 113 individuals from multi-generational Adolescent Idiopathic Scoliosis (AIS) families revealing 2566 variants, with an 18% match to known variants.
The authors managed to sequence a good number of patients, with an impressive follow-up of families, the design of the study is good, and the sequencing was done carefully. But although they count on this extremely good cohort and dataset, the manuscript lacks a conclusion and feels like a data dump where the authors did not bother to analyze anything.
Major points:
- The abstract needs to be modified, lines 15-16 can be deleted because they do not add anything.
- The introduction is very difficult to follow for anybody outside the field, between lines 33-42 the different theories for the development of AIS are not explained, and we see a dump of citations that without going through them you do not know what is going on. I would advise modifying this to make it clearer to the audience of an unspecialized journal.
- Following that paragraph, we have several citations of studies on zebrafish that are not properly linked to the previous paragraph. I would advise editing together to make the introduction easy to follow without citations that almost feel out of pocket.
- The methods section should allow anybody to replicate the study, I would suggest the authors provide us with more details on the extraction methods used, how the libraries were made, and all the technical details that we would need.
- In lines 83-85, the authors basically admit that they are slicing their project into several papers, which is more common than it should be. The issue with this is that their dataset is extremely interesting for trying to understand the penetrance of this disease. I know that functional studies take time and must be carefully designed, but it is possible to process this data a bit more to try to analyze some of the families and get candidate modulators of the disease or novel causal genes. I would encourage the authors to go over some of the families and try to add some value to the paper, as they have in their hands a very good dataset. This would be the most interesting part of the article.
- Figure 1. The pedigrees are impressive, but one can just wonder why there is no in-depth analysis of the families, even if it is just one.
- Figure 2 does not discuss it, we do not know why the authors plotted the number of genes with mutations by chromosome, is there any correlation or any interesting information that they could bring to the table with this figure?
- Figures titles and descriptions should be conclusive so we can get information from them, not just a description of the type of plot. Figure 4 could probably be simplified and edited so that the letter is readable without zooming a lot.
- The conclusion is not a conclusion.
- With such a good cohort I wonder if the authors could add clinical details and maybe some genotype-phenotype analysis. It would make the article way more interesting.
Minor points:
- Line 77, why is the COVID-19 crisis mentioned? It does not add anything to the methods.
- There are two paragraphs with the same citations, maybe the authors can modify this a bit.
Author Response
Thank you for the time spent reviewing our manuscript and the constructive comments. We have enhanced the manuscript with the findings of the familial analysis on two families as suggested, which we have found useful to improve our manuscript, thank you. You will find below the detailed response to your comments.
Major points:
- The abstract needs to be modified, lines 15-16 can be deleted because they do not add anything.
This sentence has been removed and replaced to a sentence relative to the BICD2 variants found in two families via the familial analysis of the cohort.
- The introduction is very difficult to follow for anybody outside the field, between lines 33-42 the different theories for the development of AIS are not explained, and we see a dump of citations that without going through them you do not know what is going on. I would advise modifying this to make it clearer to the audience of an unspecialized journal.
Thank you, the introduction was indeed hard to follow, and it has be rewritten
- Following that paragraph, we have several citations of studies on zebrafish that are not properly linked to the previous paragraph. I would advise editing together to make the introduction easy to follow without citations that almost feel out of pocket.
Indeed, the relevance of that section on Zebrafish here was debatable and the introduction has been edited.
- The methods section should allow anybody to replicate the study, I would suggest the authors provide us with more details on the extraction methods used, how the libraries were made, and all the technical details that we would need.
Thank you for this request, it is indeed important to report the Methods extensively, and the section has been developed.
- In lines 83-85, the authors basically admit that they are slicing their project into several papers, which is more common than it should be. The issue with this is that their dataset is extremely interesting for trying to understand the penetrance of this disease. I know that functional studies take time and must be carefully designed, but it is possible to process this data a bit more to try to analyze some of the families and get candidate modulators of the disease or novel causal genes. I would encourage the authors to go over some of the families and try to add some value to the paper, as they have in their hands a very good dataset. This would be the most interesting part of the article.
Indeed, if we are awaiting functional validation for some variants with animal models (that takes months), we have added the results regarding BICD2, found in two families after in family analysis (section 3.3 Familial analysis – BICD2 gene)
- Figure 1. The pedigrees are impressive, but one can just wonder why there is no in-depth analysis of the families, even if it is just one.
We have added an in-depth analysis of 2 families, that led us to the discovery of two variant in a gene of interest: BICD2 (3.3 Familial analysis – BICD2 gene).
- Figure 2 does not discuss it, we do not know why the authors plotted the number of genes with mutations by chromosome, is there any correlation or any interesting information that they could bring to the table with this figure?
Indeed, this figure and the repartition of the genes on the chromosomes does not bring interesting data and it has been suppressed.
- Figures titles and descriptions should be conclusive so we can get information from them, not just a description of the type of plot. Figure 4 could probably be simplified and edited so that the letter is readable without zooming a lot.
The legends of the figures have been edited accordingly.
- The conclusion is not a conclusion.
The conclusion has been rewritten.
- With such a good cohort I wonder if the authors could add clinical details and maybe some genotype-phenotype analysis. It would make the article way more interesting.
Within the familial analysis in families, we have add clinical details, as well as radiographs.
Minor points:
- Line 77, why is the COVID-19 crisis mentioned? It does not add anything to the methods.
Indeed, we have removed the mention to Covid crisis
- There are two paragraphs with the same citations, maybe the authors can modify this a bit.
The introduction has been rewritten in that section.

Reviewer 2 Report
Comments and Suggestions for Authors
The Authors provide a study about genetic vsriants in AIS
Abstract >> clear; ## Introduction >>ok
Mat and meth >>
Line 68, erase bracketts after gibbosity; # line 68-69 check language
Line 117: French citizens : it is not informative, many French citizens are not Caucasians, so please specify better ethnicity. See also other comments below.
Line 123 and fig.2 The distribution of variants /chromosome likely is random; if not, it should be explained in methods why this calculation was done; consider erasing it unless a good rationale exists.
Discussion
Line 180/181 specify ethnicity
Line 194/195/196 the statement is correct; you may add a comment about incidence of AIS in different populations and a related comment about ethnicity of your families
Line 211/212 check: sentence not clear
Line 224 specify if the cohort is comparable just in size or also in ethnicity
Line 232 I understand that in 2023 it is not possible not to quote epigenetic studies (!), but is there any relation between genes epigenetically modified and the variant containing genes that you identified? Please comment.
Comments on the Quality of English LanguageEnglish good enough, just some minor changes could be made;
check the sentences as indicated before
Author Response
Thank you very much for the time spent to review our article. We understand that ethnicity is an important component in genetic study, but unfortunately, unlike in other countries, French law do not allow doctors to routinely collect data on patients ethnicity. We have carried
Abstract >> clear; ## Introduction >>ok
Mat and meth >>
Line 68, erase bracketts after gibbosity. Thank you for your vigilance, done.
# line 68-69 check language: Thank you, language has been improved.
Line 117: French citizens : it is not informative, many French citizens are not Caucasians, so please specify better ethnicity. See also other comments below.
This is indeed an interesting comment. Unfortunately, French law does not allow to collect data regarding race routinely and the Ethical Approvement we have obtained does not comprised race in the observational data sheet. For your information, most families were Caucasian; there were no Asiatic or African family, and three families were with an Israeli background.
Line 123 and fig.2 The distribution of variants /chromosome likely is random; if not, it should be explained in methods why this calculation was done; consider erasing it unless a good rationale exists.
Indeed, this figure and the repartition of the genes on the chromosomes does not bring interesting data and it has been suppressed.
Discussion
Line 180/181 specify ethnicity
Line 194/195/196 the statement is correct; you may add a comment about incidence of AIS in different populations and a related comment about ethnicity of your families
This is indeed an interesting comment. Unfortunately, French law does not allow to collect data regarding race routinely and the Ethical Approvement we have obtained does not comprised race in the observational data sheet. For your information, most families were Caucasian; there were no Asiatic or African family, and three families are with an Israeli background.
Line 211/212 check: sentence not clear
The sentence wasn’t clear indeed, and the section has been edited, thank you.
Line 224 specify if the cohort is comparable just in size or also in ethnicity
This is indeed an interesting comment. Unfortunately, French law does not allow to collect data regarding race routinely and the Ethical Approvement we have obtained does not comprised race in the observational data sheet. For your information, most families were Caucasian; there were no Asiatic or African family, and three families are with an Israeli background.
Line 232 I understand that in 2023 it is not possible not to quote epigenetic studies (!), but is there any relation between genes epigenetically modified and the variant containing genes that you identified? Please comment.
On the nine genes epigenetically modified retained as the most significant by the cited study of Faldini et al., none of them is found in our list of variants containing genes. Unfortunately, there is no clear correlation between epigenetic data in the literature and our results. However, we indeed thought (as you said) that epigenetic was to mention in the discussion as its involvement in AIS is proven.
Round 2
Reviewer 1 Report
Comments and Suggestions for Authors
I congratulate the authors for improving the manuscript and going in the right direction, but there are still things that need a slight improvement.
The methods are now more understandable and there are very interesting new results on two families, the proteins fit the story and I am sure functional assays will be extremely valuable.
If possible, I would prefer the track changes instead of just having everything in yellow and having to compare with the previous version, which is a waste of time.
Major points:
- The introduction is still difficult to comprehend for a wide audience, just spend a couple of lines explaining the different theories so the readers do not have to go back to the citations over and over.
- Figure 3 is still painful to see, authors should think of splitting it in different panels or building on it differently. Readers have to be able to see what is important in the figure by just looking at it. Maybe other tools like String could be useful to prioritize a bit.
- Can the authors carry out any phenotype-genotype association? I know they have the data and new candidate genes, is there anything that they can include in the article to increase the value?
Minor:
- Check some small errata like some double points in lines 261 and 277.
Out of curiosity, what functional assays are the authors planning?
Author Response
"I congratulate the authors for improving the manuscript and going in the right direction, but there are still things that need a slight improvement. The methods are now more understandable and there are very interesting new results on two families, the proteins fit the story and I am sure functional assays will be extremely valuable.
ïƒ Thank you very much.
If possible, I would prefer the track changes instead of just having everything in yellow and having to compare with the previous version, which is a waste of time.
ïƒ We have implemented the changes in revision mode in the word file.
Major points:
- The introduction is still difficult to comprehend for a wide audience, just spend a couple of lines explaining the different theories so the readers do not have to go back to the citations over and over.
ïƒ We have expanded the introduction to add some details on the different theories and the genes, for it not to appear as a list of citations.
- Figure 3 is still painful to see, authors should think of splitting it in different panels or building on it differently. Readers have to be able to see what is important in the figure by just looking at it. Maybe other tools like String could be useful to prioritize a bit.
ïƒ We have added on the figure a visual pooling of the different categories, that we hope will make it easier to comprehend. String was not able to support the GO term list given by BINGO and we wanted to stick with the original data, so readers that are interested can zoom in to the categories, and the other readers can understand its purpose by looking at it. Thank you for your remark.
- Can the authors carry out any phenotype-genotype association? I know they have the data and new candidate genes, is there anything that they can include in the article to increase the value?
ïƒ There is no clinical feature of muscular impairment in the two BICD2 families. We have detailed that point in the text. Unfortunately, they did not consent for an EMG, which would have been interesting to demask infra-clinical features. Regarding the curves, they are mainly right thoracic curves, with an age of onset not different from other families.
Minor:
- Check some small errata like some double points in lines 261 and 277.
ïƒ Thank you for your vigilance, the double points have been removed.
Out of curiosity, what functional assays are the authors planning?
ïƒ We are raising a Zebrafish animal model. We hope to publish those results once the animals will be ready for phenotypic analysis.